# Anti-Apoptotic Effect of Flavokawain A on Ochratoxin-A-Induced Endothelial Cell Injury by Attenuation of Oxidative Stress via PI3K/AKT-Mediated Nrf2 Signaling Cascade

**DOI:** 10.3390/toxins13110745

**Published:** 2021-10-21

**Authors:** Peramaiyan Rajendran, Abdullah M. Alzahrani, Vishnu Priya Veeraraghavan, Emad A. Ahmed

**Affiliations:** 1Department of Biological Sciences, College of Science, King Faisal University, Al Ahsa 31982, Saudi Arabia; aalzahra@kfu.edu.sa (A.M.A.); eaahmed@kfu.edu.sa (E.A.A.); 2Department of Biochemistry, Saveetha Institute of Medical and Technical Sciences, Saveetha Dental College, Saveetha University, Chennai 600077, India; vishnupriya@saveetha.com; 3Laboratory of Molecular Physiology, Zoology Department, Faculty of Science, Assiut University, Assiut 71515, Egypt

**Keywords:** flavokawain A, ochratoxin A, oxidative stress, apoptosis, Nrf2

## Abstract

This study investigates the endothelial protective activity of flavokawain A (FKA) against oxidative stress induced by ochratoxin A (OTA), which acts as a mycotoxin, and its primary mechanisms in in vitro models. Reactive oxygen species, in general, regulate oxidative stress that significantly contributes to the pathophysiology of endothelial dysfunctions. OTA exerts toxicity through inflammation and the accumulation of ROS. This research is aimed at exploring the defensive function of FKA against the endothelial injury triggered by OTA through the Nrf2 pathway regulated by PI3K/AKT. OTA exposure significantly increased the nuclear translocation of NFκB, whereas we found a reduction in inflammation via NFκB inhibition with FKA treatment. FKA increased the PI3K and AKT phosphorylation, which may lead to the stimulation of antioxidative and antiapoptotic signaling in HUVECs. It also upregulated the phosphorylation of Nrf2 and a concomitant expression of antioxidant genes, such as HO-1, NQO-1, and γGCLC, depending on the dose under the oxidative stress triggered by OTA. Knockdown of Nrf2 through small interfering RNA (siRNA) impedes the protective role of FKA against the endothelial toxicity induced by OTA. In addition, FKA enhanced Bcl2 activation while suppressing apoptosis marker proteins. Therefore, FKA is regarded as a potential agent against endothelial oxidative stress caused by the deterioration of the endothelium. The research findings showed that FKA plays a key role in activating the p-PI3K/p-AKT and Nrf2 signaling pathways, while suppressing caspase-dependent apoptosis.

## 1. Introduction

The endothelium helps provide appropriate hemostatic balance. Vascular endothelial cells constitute the inner cellular lining of the circulatory system. The unique functions of these cells are important to the vascular biology that deals with hemostasis, regulation of blood vessel tone, kidney glomeruli functions such as fluid filtration, trafficking of hormone and neutrophil. According to researchers in this field, the endothelium represents various homeostatic functions [1,2,3,4]. The main pathophysiological mechanisms of a number of diseases are endothelial dysfunction and oxidative stress, including hypertension, atherosclerosis, dyslipidemia, diabetes, cardiovascular disease, renal failure, and ischemia-reperfusion injury [5,6,7,8]. When present in excess, reactive oxygen species (ROS) modulate cellular function, receptor signals, and immune responses, causing progressive endothelial damage through growth and migration of vascular smooth muscle and inflammation cells. The extracellular matrix is then altered, endothelial cells are apoptosed, transcription factors (NFkB, AP-1) are activated, and inflammatory cytokines are overexpressed [9]. Increased oxidative stress in vascular tissues is caused by ROS-producing enzymes, such as NADPH oxidase (Nox), xanthine oxidase, and the mitochondrial respiratory chain. Several superoxide anions are known to react directly with NO by producing peroxynitrite (ONOO*), which is believed to uncouple endothelial nitric oxide synthase (eNOS) and cause an NO-producing enzyme to become an ROS-producing enzyme, thus accelerating the atherosclerotic process [10].

Ochratoxin A (OTA) is a powerful mycotoxin found in many foods and feed. It is responsible for both chronic and sub-chronic toxicity, including nephrotoxicity, hepatotoxicity, teratogenicity, and immunotoxicity to both humans and various animals [11]. Chronic toxicity is one of the main causes of OTA, resulting from its prolonged intake in minimum amounts. OTA associates itself with hepatotoxic, genotoxic, carcinogenic, nephrotoxic, teratogenic, and immunosuppressive effects, affecting humans as well as various animals [12,13]. The effects of OTA are widely known in the research community; however, researchers have not completely clarified the molecular mechanisms underlying the damage. However, there is currently no definitive line of evidence for endothelial dysfunction caused by OTA exposure. In the present study, we evaluate endothelial dysfunction caused by OTA through oxidative stress and the protection against oxidative stress induced by OTA in endothelial cells by using natural compounds. OTA exposure (in vitro or in vivo) has generally been linked to oxidative damage (lipids, proteins, and DNA) and the overproduction of ROS [14]. This particular mycotoxin may also decrease the antioxidant defense in cells by reducing GSH and cytoprotective enzymes [11]. Furthermore, it significantly reduces the expression of genes that consist of antioxidant regulatory elements (AREs) and causes oxidative stress in the kidney tissues. Transcription factors, such as NF-E2-related factor 2 (NRF2), identify specific elements, including AREs, and contribute to the regulation of antioxidant enzymes, cell protection, and genes encoding detoxification [15]. The PI3K/AKT pathway also provides cellular defense against oxidative stress and inflammatory stimuli. According to many studies, various signal transduction mechanisms, including PI3K/AKT, influence Nrf2 to detach from Keap1 and provide successive signals to trigger activation of antioxidant enzymes [15]. These studies have also proved that oxidative stress results in the decreased regulation of PI3K/Akt signaling for cell survival. Therefore, regulating the routes of signaling pathways, such as PI3K/Ak,t is a potential way to prevent hepatic apoptosis induced by ROS [16]. Subsequently, the inhibition of oxidative stress mediated by OTA can emerge as a potential therapeutic strategy for vascular dysfunction treatment. Numerous studies have investigated the endothelial protective effects of natural compounds against liver injury induced by toxic chemicals, which have been attributed to the intrinsic antioxidant properties [8,11,15,17]. Recently, Zhai et al. demonstrated that a dietary supplementation of curcuminoid composition (CURC) not only reversed the biochemical changes in serum but also ameliorated liver oxidative injury in OTA-treated white Pekin ducklings (for 3 weeks) [17].

Flavokawain A (FKA), also called kava–kava, is a natural chalcone derived from *Piper methysticum* Forst [18]. Age-standardized cancer cases in three kava-consuming Pacific countries (Fiji, Samoa, and Vanuatu) were reported to be significantly lower when compared to those of their neighboring countries (Australia and New Zealand). Multiple secondary metabolites, such as chalcones (flavokawains), lactones (kavalactones), and alkaloids in kava extracts, contribute to their therapeutic actions. Until now, researchers have identified only three kinds of flavokawains (FKA, FKB, and FKC). Among these types, FKA is considered the predominant chalcone, accounting for about 0.46% of kava extracts [19]. According to earlier studies, FKA selectively inhibits cancer cell growth with a zero or low effect on the progression of various tumor cells [20,21]. Dietary feeding of mice with FKA had no negative effects on the key functions of the organs; instead, it generated phase II antioxidant enzymes in the lung, liver, bladder, and prostate tissues [22]. However, researchers have not yet investigated FKA effects on damages to the vascular system and apoptosis-related signaling molecules. In the current study, we propose the hypothesis that FKA can mitigate oxidative stress triggered by OTA and Nrf2 signaling mediated by PI3K/AKT, which, in turn, will prevent endothelial dysfunction in HUVECs. We also found the important molecular proteins that demonstrate the anti-apoptotic role of FKA and activate Nrf2 signaling on OTA induced endothelial damage.

## 2. Results

### 2.1. Effects of FKA on the Viability of HUVECs with or without OTA Stimulation

Before exploring the structure of FKA (Figure 1A), we fixed the effective concentration of OTA at 10 µmol based on the dose–response (Figure 1B). HUVECs were treated with rising FKA concentrations (0–50 µmol) for 24 h to determine the cytotoxic effect on these cells. Based on the results of the MTT assay, the viability of HUVECs was the same (100%) following the treatment with FKA, indicating that, up to 50 µmol, FKA did not generate any cytotoxic effects (Figure 1C). This was followed by a further evaluation of the impact of FKA (0–25 µmol) on the cell viability and morphology of HUVECs with or without OTA (10 µmol) induction. Cell pre-treatment with FKA (0–25 µmol) strongly suppressed the OTA-induced (10 µmol) viability of HUVECs (Figure 1D). Images obtained from optical microscope did not reveal any morphological changes after FKA treatment (Figure 1E). These findings indicated that FKA had no cytotoxic effects on HUVECs.

### 2.2. Effect of FKA on the Activation of ROS-Mediated NFκB and Pro-Inflammatory Cytokines against OTA-Induced HUVECs

The transcription factor NFκB regulates various factors of innate and adaptive immune functions. It also acts as a key mediator of the response to inflammation [23]. In this experiment, we examined NFκB function in suppressing inflammation in the endothelium, which was pretreated with FKA and stimulated by OTA. Data from Western blot collected from the extracts of the nuclear protein of HUVECs illustrated that OTA (10 μmol) stimulation triggered a significant rise in the levels of p65 (Figure 2A). Further, we examined the effects of FKA on pro-inflammatory cytokine such as Cox-2 and TNF-α expression. Data generated from Western blot also demonstrated that OTA (10 μmol) stimulation overexpressed the COX-2 and TNF-α expressions (Figure 2B). However, the presence of FKA in higher concentrations (25 μmol) significantly suppressed this effect. It consequently suggested that the transcriptional activation of p65 was suppressed by FKA, which further suppressed the expressions of inflammatory enzymes and cytokines. To reconfirm our findings, we tested secretion pro-inflammatory cytokines using ELISA. It was observed that OTA stimulation only enhanced the production of TNF-α, IL-6, and IL-1β cytokines (Figure 2C–E). However, the pretreatment with FKA remarkably suppressed the production of pro-inflammatory cytokines in a dose-dependent manner, thereby indicating the defensive action of FKA against the inflammation in HUVECs triggered by OTA.

### 2.3. FKA Protects the Endothelium against OTA-Induced Apoptosis

Apoptosis remains a crucial indicator of OTA-induced vascular toxicity, which can also be worsened by uncontrolled oxidative stress and inflammation. We directly analyzed the anti-apoptotic capacity of FKA through terminal deoxynucleotidyl transferase dUTP nick-end labeling (TUNEL) staining, showing reduced TUNEL-positive nuclei in OTA-treated cells (Figure 3A,B). These results showed that FKA inhibited OTA-induced apoptosis. The expression of apoptosis-related proteins, such as Bcl-2, demonstrated OTA-induced apoptosis in HUVECs via potential downregulation of protein expression levels (Figure 3C). FKA treatment, on the other hand, upregulated the proteins for cell survival, thereby preventing apoptosis. Our data from Western blot analysis also demonstrated changes in the expression of cleaved caspase-3 and cleaved PARP—the two proteins associated with apoptosis. Although OTA upregulated the expression of these proteins, different concentrations of FKA downregulated the expression levels (Figure 3C).

### 2.4. FKA Inhibits Apoptosis in HUVEC Line

DNA fragmentation is one of the characteristic hallmarks of apoptosis. As mentioned above, we examined DNA fragmentation using a cell death detection ELISA kit [8]. OTA-treated cells saw a significantly increased DNA fragmentation, whereas FKA generated significant 5- and 7-fold decreases in DNA fragmentation in HUVECs (Figure 3D), following treatment for 24 and 48 h, respectively, in comparison to the cells treated with the combination of FKA and OTA. This indicated that FKA was a potent inhibitor of apoptosis in OTA-induced oxidative stress in vascular endothelial cells.

### 2.5. Effect of FKA on ROS Generation against OTA-Induced Endothelial Cells

The activation of Nrf2 regulates various signaling cascades, such as ROS. These are considered as natural oxygen metabolism by-products and play a key role in homeostasis and cell signaling. Low concentrations of ROS play an indispensable role in intracellular signaling and defense against pathogens, while higher amounts of ROS play a role in a number of human diseases [24]. Like other chalcones, FKA demonstrated effective antioxidant and chemo-protective properties in various experiments, in vitro as well as in vivo [25,26]. In an earlier study, we showed the protective role of FKA in vascular HUVECs in a case of oxidative stress induced by TGF-β1 (10 ng/mL) [18]. In this study, we investigated the effect of ROS generation induced by OTA on HUVECs. We used the method of DCFH2-DA fluorescence to measure the intracellular generation of ROS. Endothelial cells were subjected to FKA pretreatment in varying concentrations (0, 10, and 25 μmol for 2 h) followed by OTA induction (10 μmol) for 24 h (Figure 4A). We then used these cells to calculate the levels of intracellular ROS accumulation. Data signified that dose-dependent FKA pretreatment remarkably suppressed ROS generation triggered by OTA, thereby confirming its role in blocking ROS generation in HUVECs.

### 2.6. FKA Upregulates Nrf2, HO-1, and γ-GCLC in HUVECs

According to earlier studies, Nrf2 is involved in the anti-inflammatory process, as it improves inflammatory cell recruitment and enables gene expression regulation AREs. Keap-1/Nrf2/ARE is primarily an antioxidant pathway and regulates anti-inflammatory gene expression, which further suppresses inflammation progression [27]. Endothelial cells were pretreated with different FKA concentrations (0, 5, 10, and 25 μmol for 2 h) and inducted with OTA (10 μmol) for 24 h. We also measured several antioxidant protein expressions. Western blot data obtained from this study demonstrated that dose-dependent FKA treatment upregulated the nuclear Nrf2, HO-1, NQO-1, and γ-GCLC expressions, and we observed the maximum expressions observed at 25 μmol (Figure 4B). We measured the data as the fold over basal levels of the expression of antioxidant proteins at varying dose points. This was followed by the normalization of these points using the histone and β-actin internal controls, which are expressed as the ratio value. Data revealed a differential yet substantial FKA effect on the nuclear Nrf2, HO-1, γ-GCLC, and NQO-1 expression patterns.

### 2.7. Effect of FKA on OTA-Induced GSH Levels in Endothelial Cells

FKA is considered to play a remarkable role in triggering antioxidant proteins. Consequently, we measured the GSH levels of endothelial cells pretreated with different FKA concentrations (0, 5, 10, and 25  μmol for 2  h) and inducted with OTA (10 μmol) for 24  h. FKA treatment is predominantly responsible for restoration of the levels of intracellular GSH (Figure 4C). GSH increase could be achieved via FKA-mediated direct ROS scavenging and through the upregulation of γ-GCLC that is involved in GSH synthesis.

### 2.8. Effects of FKA on Nrf2-Related mRNA Expression in OTA-Induced Endothelial Cells

Considering the role of FKA in stimulating the pathway of Nrf2/HO–1/NQO1, we analyzed whether FKA was involved in the Nrf2/HO–1/NQO1 pathway activation in OTA-supplemented cells. The qRT-PCR results showed significant upregulation of mRNA (*p* < 0.05) in FKA-pretreated cells, in contrast to those treated only with OTA (Figure 4D). According to these data, FKA upregulated Nrf2 and subsequently ameliorated OTA-triggered oxidative stress.

### 2.9. FKA Activates OTA-Induced PI3K and AKT Phosphorylation in HUVECs

The inhibitors of pPI3K/pAKT increase the risk of vascular inflammation and endothelial dysfunction [28]. Therefore, we further checked the role of FKA in inhibiting the OTA-triggered oxidative stress via a signaling pathway such as PI3K/AKT using Western blotting. As shown in Figure 5A, we observed that OTA alone (10 µmol) could downregulate the activation of pPI3K/pAKT. However, after treating the cells with FKA and OTA, the pPI3K/pAKT activation was remarkably and dose-dependently upregulated. This finding suggested the protective role of FKA against OTA-induced oxidative stress via a signaling pathway such as PI3K/AKT.

### 2.10. Knockdown of Nrf2 (siRNA) Attenuated the Protective Effect of FKA on HUVECs under Oxidative Stress

To further validate that FKA-induced Nrf2 plays a primary role in oxidative stress, Nrf2 was knocked down by siRNA transfection of Nrf2, and HO-1, γ-GCLC, and NQO-1 expressions were subsequently assayed. Western blot analysis suggested that cells that were Nrf2 knockdown and exposed to FKA (25 μM) showed decreased γ-GCLC, NQO-1, and HO-1 expression (Figure 5B). This additional evidence affirmed that the FKA-activated Nrf2 signaling promoted diminished oxidative stress against OTA-induced vascular damages in HUVECs.

### 2.11. FKA Treatment Inhibits Apoptosis-Related Morphological Changes in HUVECs

To observe if FKA treatment induced oxidative stress as a source of morphological changes associated with apoptosis, we conducted AO/EB-based dual staining. Controls and cells treated with DMSO became dark green, showing the cells’ viability. It is worth noting that OTA-treated HUVECs strongly triggered the viable cells to shift to EB-positive late or dead apoptotic cells. FKA treatment also resulted in a concentration-dependent inhibition of the EB-positive apoptotic cells (Figure 6A). The result suggested that a decrease in apoptotic cell population after FKA treatment was markedly concentration-dependent.

### 2.12. FKA Activates PI3K/AKT Signaling to Regulate Nrf2 in OTA-Induced Endothelial Cells

To examine whether the upregulation of pAKT was related to Nrf2 signals, we investigated Akt protein stimulation using Western blotting. In this experiment, we used a selective inhibitor, LY294002, to determine if PI3K/AKT played an important role in activating Nrf2 signaling in cells pretreated with FKA and OTA. As shown in Figure 6B, cells pretreated with the combination of OTA and FKA upregulated pAKT, Nrf2, and HO-1. On the other hand, LY294002-treated cells, and the combination of FKA and OTA significantly downregulated the levels of these proteins. Therefore, it was affirmed that FKA may depend on the PI3K/AKT–Nrf2 signaling regulation in HUVECs.

## 3. Discussion

The toxicity and common contamination by OTA in food and feed have garnered significant attention in the food safety and animal feed contamination sectors. There has also been a growing demand to discover effective pathways to limit the toxicity of OTA. Moreover, microbiology, toxicology, and food technology professionals have been paying special attention to OTA [11]. Various published in vitro and in vivo studies have focused on OTA nephrotoxicity and hepatotoxicity. However, the underlying mechanisms and the impact of oxidative stress on the harmful effects of these mycotoxins are not yet clear. Many researchers have studied various antioxidant compounds for their protective effects against OTA-induced organ toxicity [29,30]. However, there is no research on the effective protective effects of FKA against the endothelial toxicity induced by OTA. Recent studies have focused on the biological impacts of FKA for its benefits, such as anti-inflammation and antioxidant properties. Consequently, the current study explored the protective mechanism of FKA against OTA-triggered endothelial apoptosis.

For this study, we stimulated HUVECs with OTA to trigger an inflammatory reaction and subsequently demonstrated the FKA effect. According to our data, 10 μmol of OTA dramatically upregulated cytokines, such as TNF-α, IL-1β, and IL-6. The pretreatment of cells with FKA, however, contributed to significant downregulation of TNF-α, IL-1β, and IL-6 secretions at all concentrations. This result indicated that FKA serves as a possible anti-inflammatory agent protecting the endothelium against inflammation.

According to recent evidence, ROS take part in inflammatory reactions. Additionally, ROS production triggered by OTA has been studied in a wide range of in vitro and in vivo systems [31,32,33]. For this study, it is important to note that ROS are oxidative products of the peroxisomes, mitochondria, and inflammatory cell activation of endotoxins in alveolar macrophages, and cytochrome p450 metabolism [34,35]. In concordance with the literature, our flow cytometry study data suggested that the treatment of cells with OTA alone stimulated the dramatic upregulation of the ROS levels in HUVECs. However, dose-dependent FKA pretreatment significantly suppressed the previous effect. Accordingly, the Western blot data showed significant upregulation of the nuclear p65, COX-2, and TNF-α expressions due to OTA induction. Nevertheless, the pretreatment with FKA substantially suppressed this effect, indicating blocking of the transcriptional activation of p65, resulting in the downstream suppression of the expressions of inflammatory enzymes and cytokine proteins.

In the growth factor superfamily, PI3K is considered a key signal transduction molecule. With the help of kinases that depend on P13K, the serine and threonine residues of P13K phosphorylation contribute to the activation of AKT. According to various reports, PI3K/AKT signaling enables the regulation of ROS expression and the pathways of cellular oxidative stress [8]. It presents a survival signal, allowing Keap-1 to produce Nrf2 and its subsequent translocation, and further regulates the activation of Nrf2 that depends on ROS under various stresses, including oxidative stress. Conversely, a few researchers have suggested that some kinase pathways, such as PI3K, can restore the Nrf2–ARE activation without depending on oxidative stress [18]. To assess if FKA upregulates HO-1 via the activation of Nrf2, we isolated the protein and inspected the expression of Nrf2 protein using Western blotting. We observed that the supplementation of FKA remarkably augmented the expression of Nrf2 protein in the nucleus, suggesting its key role in stimulating the expression of γGCLC and HO-1 through Nrf2 signaling regulation. Furthermore, we used cells knocked down by Nrf2 to learn whether the inhibition of signaling regulated by Nrf2 improved FKA-induced HO-1 and γGCLC expression. The inhibition of Nrf2 was observed to profoundly attenuate the HO-1 expression triggered by FKA. The results thus suggested that FKA exhibits antioxidant properties through the upregulation of the expression of HO-1 via Nrf2 signaling.

We proposed a hypothesis that PI3K/AKT plays a role in regulating FKA in cells treated with OTA. FKA supplementation dramatically increased the AKT protein phosphorylation, indicating that increased phosphorylation of the AKT protein may be involved in the expression of HO-1 stimulated by FKA. To assess the PI3K/AKT pathway function, we used a particular inhibitor of PI3K/AKT, LY294002 for HUVECs, and a combination of FKA and OTA. The findings showed that FKA induced PI3K/AKT and Nrf2 expressions. Additionally, we investigated whether PI3K/AKT played a key role in limiting the generation of ROS via FKA. Our findings indicated that the cascade of PI3K/AKT signaling plays a crucial role in FKA-triggered HO-1 expression by inducing Nrf2 in oxidative stress induced by OTA.

The stimulation of endothelial injury via OTA-induced oxidative stress results in cell apoptosis. Caspase-3 and Bcl-2 are essential proteins that stimulate the pathway of apoptotic cell death [16]. In this study, we examined the FKA mechanisms against cytotoxicity triggered by OTA. We observed that FKA slightly reversed the alteration of OTA-mediated proteins in Bcl-2, cleaved caspase-3, and cleaved PARP. Moreover, OTA can lead to HUVEC death by provoking the apoptosis of cells. Therefore, it is considered that FKA provides protection against damage caused by OTA through the blockage of the apoptotic cell death pathway. FKA may further improve the pathway of DNA repair.

## 4. Conclusions

Dysfunction of the vascular endothelium is thus a hallmark of human diseases. In this respect, the early detection and immediate treatment of endothelial dysfunction appear crucial for substantial recovery from endothelium-caused diseases. Our study findings validated that oxidative stress contributes to the mechanism of OTA endothelial toxicity, and exposure to OTA triggers adverse effects and significant transformation of endothelium functions. FKA treatment contributes to restoration of endothelium function through PI3K/AKT-mediated Nrf2 signaling. Therefore, the use of FKA for its antioxidant activity can suppress oxidative stress and decrease the biosynthesis of mycotoxins in food sources while protecting human and animal health.

## 5. Materials and Methods

### 5.1. Chemicals

We bought both OTA and FKA from LKT Labs Inc. (St Paul, MN, USA). OTA and FKA were dissolved in 0.1 mol/L sodium bicarbonate and DMSO, respectively. For this study, we also used other analytical grade reagents.

### 5.2. Cell Culture

HUVECs were collected from the American Type Culture Collection (ATCC, Manassas, VA, USA). The cells procured were developed using ECM basal medium enriched with 10% fetal bovine serum (ThermoFisher Scientific, Waltham, MA, USA), 2 mM glutamine (Sigma–Aldrich, St. Louis, MO, USA), 100 unit/mL penicillin (Sigma–Aldrich), 100 μg/mL streptomycin (Sigma–Aldrich), and 1 mM pyruvate (Sigma–Aldrich) in humidified air (5% CO_2_) at 37 °C.

### 5.3. Cell Culture and Treatment

We cultured the cells in a 37 °C incubator with CO_2_ of 5% and passaged them 3–5 times before using them in the experiments. After the cells were grown to 90% confluence, they were inoculated in 6- and 96-well plates. We pretreated the endothelial cells with different FKA concentrations (0, 10, and 25 μmol for 2 h) and inducted OTA (10 μmol) for 24  h.

### 5.4. In Vitro Stimulation Assays

In this experiment, we pretreated the HUVEC cultures for 2 h with FKA. The cells washed with phosphate-buffered saline were then subjected after incubation to a new medium supplemented with or without OTA (prepared in 0.1 mol/L NaHCO_3_). Li Zhang et al.’s (2019) study showed inflammation in HUVECs using 12.5 ng/mL TNF-α for 12 h [36]. Based on this experiment, 10 μmol of OTA was used for HUVECs.

To calculate TNF-α, IL-1β, and IL-6 levels in the cell culture medium, about 6.5 × 105 cells/well of HUVECs were cultured in a 12-well plate. The cells were pretreated with FKA (0–25 μM, 2 h) followed by OTA (10 μmol, 72 h), and we used the ELISA method for measuring the former’s protective effect against the secretion of cytokine from cells stimulated by OTA. The ELISA kits (R&D Systems, Minneapolis, MN, USA) quantified their respective cytokines according to the manufacturer’s protocols.

### 5.5. Estimation of Total Glutathione

The GSH content was determined using the Glutathione Assay Kit (Catalog No. CS0260; Sigma–Aldrich Inc.). The samples from each treatment were cleaned with 0.9% NaCl. The clean samples were homogenized in trichloroacetic acid (1:4, *w*/*v*) using a Teflon homogenizer and centrifuged at 3000× *g* and 4 °C for 10 min. The supernatant was collected, and the GSH content of the supernatant was measured at 420 nm according to the manufacturer’s protocol using the Varioskan Flash spectrophotometer (ThermoFisher Scientific). For measuring the total GSH content, standard curves were obtained with GSH equivalents of 0, 150, and 350 μM. [37].

### 5.6. Western Blotting

Post-treatment, we harvested the cells and used cold PBS to wash them. We then prepared nuclear, cytoplasmic, and total extracts in the aforementioned manner. For detecting the status of the protein, we used a Bio-Rad protein assay in each sample, with bovine serum albumin (BSA) as the reference standard. To obtain protein (50 μg) in equal amounts, we used SDS-PAGE (8–15%) and transferred the proteins to nitrocellulose membranes overnight. We blocked the membranes using 5% skimmed milk at 3 °C for 30 min and then incubated them for 2 h with the indicated primary antibodies (1:1000 dilution). Subsequently, a horseradish-peroxidase-conjugated goat anti-mouse or anti-rabbit secondary antibody (1:5000 dilution) was incubated using the nitrocellulose membranes for 1 h. Importantly, we used an improved chemiluminescence substrate (Pierce Biotechnology, Rockford, IL, USA) for membrane development.

### 5.7. Measurement of ROS Generation

In this study, we identified the generation of intracellular ROS through fluorescence microscopy using the cell-permeable fluorogenic test DCFH2-DA [38]. Cells (2.5 × 10^4^ cells/mL) were developed in 10% FBS-supplemented ECM basal medium, and when the cells reached 80% confluence, we replaced the culture medium. Post-treatment, we expelled and cultured the culture supernatants using non-fluorescent DCFH2-DA (10 μM) in a new medium at 37 °C for 30 min. The production of intracellular ROS was examined through the calculation of the intracellular amassing of dichlorofluoresce in (DCF) resulting from the oxidation of DCFH2. The fluorescence emitted was calculated using LS 5.0 delicate picture arrangement examination (Olympus Imaging America Inc., Center Valley, PA, USA).

### 5.8. DNA Fragmentation

The nuclear DNA fragmentation into nucleosomal units is a distinctive feature of programmed cell death. It is a response to different apoptotic stimuli in various types of cells. In this experiment, the DNA fragmentation in OTA and/or FKA-treated HUVECS was determined using the Cell Death Detection ELISA PLUS kit (Roche Applied Science, Branford, CT, USA) as per the manufacturer’s instructions as mentioned above [8].

### 5.9. RT-PCR

We cleaned the FKA-injected cells with PBS and used TRIzol reagent (Invitrogen, Carlsbad, CA, USA) to isolate HUVEC RNA. We then used a PrimeScript RT reagent kit to convert the RNA to cDNA, as per the manufacturer’s guidelines (Takara Bio, Shiga, Japan). We then performed real-time qPCR with the SYBR Green system (Applied Biosystems, Foster City, CA, USA) and ViiA-7 Applied Biosystem (Carlsbad, CA, USA). In all genes, the expression of mRNA was standardized to the β-actin housekeeping gene expression. We determined the status of the expression of mRNA (fold change) between groups by 2-ΔΔCt value in comparison with the non-treated (NT) samples [8].

### 5.10. Cytoplasmic and Nuclear Extractions

In this experiment, cell pellets were resuspended in Buffer I, consisting of 25 mM HEPES at pH 7.9, 5 mM KCl, 0.5 mM MgCl2, and 1 mM dithiothreitol (DTT), for 5 min for the preparation of the cytoplasmic extracts. We then mixed this suspension with an equal amount of Buffer II containing 25 mM HEPES at pH 7.9, 5 mM KCl, 0.5 mM MgCl_2_, and 1 mM DTT. Furthermore, the suspension supplemented with the inhibitors of protease and phosphatase was added to 0.4% (*v*/*v*) NP40. We incubated the suspension samples obtained from this experiment with spin at 4 °C for 15 min. The subsequent procedure involved the centrifugation of the lysates in a microfuge at 2500 rpm at 4 °C for 5 min. We then transferred the supernatants to new Eppendorf tubes. We cleaned the pellets once using Buffer II, and we added the supernatant to the cytoplasmic protein tube. For removing the residual nuclei, we centrifuged the lysates again for 5 min at 4 °C at 10,000× *g* and then emptied to new Eppendorf tubes.

For nuclear extraction, the pellets formed from the cytoplasmic extraction were incubated with Buffer III. Apart from the inhibitors of protease and phosphatase, Buffer III consisted of 25 mM HEPES, pH of 7.9, 400 mM NaCl, 10% dextrose or sucrose, 0.05% NP40, and 1 mM DTT. We rotated the lysates for 1 h at 4 °C, followed by 10-min centrifugation at 4 °C at 1000 rpm. It was observed that the collected supernatants contained nuclear proteins [8].

### 5.11. AO/EB Stain

We stained the cells treated with FKA (10 and 25 μmol) and OTA (10 μmol) using an acridine orange/ethidium bromide (AO/EB, 100 μg/mL) mixture at room temperature for 5 min. We observed the stained cells using fluorescence microscopy (Zeiss, München, Germany) at 100× magnification. We counted over 300 cells/sample in each experiment [39].

### 5.12. Transfection

We performed transfection with a 5′–3′ sequence that targets human Nrf2 siRNA. We loaded HUVECs at 1.5 × 10^5^ cells per well into 6-well plates and performed transfection with Lipofectamine 2000, following the manufacturer’s recommendations. Shortly after, we prepared the right amount of Nrf2 siRNA along with 5μL Lipofectamine 2000 in 250 μL serum-free DMEM/12 medium in individual RNase-free tubes. The 5 min incubation of siRNA and Lipofectamine was followed by the combination and incubation for another 20 min and supplementation to each well. After incubating for 24 h with 100 pM siRNA per well, FKA was added to the cells for protein analysis for 24 h [40].

### 5.13. TUNEL Assay

In the log phase, HUVECs were loaded into an FKA- or OTA-supplemented 6-well plate. After removing the medium, we cleaned the cells with phosphate buffer saline and processed them for about 20 min with 4% paraformaldehyde. This process was followed by the removal of paraformaldehyde. The cells were re-washed with phosphate buffer saline and were then subjected to incubation with TUNEL reagent (11684817910, Roche, Mannheim, Germany). We used 0.1 μg/mL DAPI to counterstain the washed cells for 5 min and studied them through a fluorescence microscope. We executed all the morphometric-related studies three times. TUNEL-positive cells were identified as brilliant green, whereas we observed the cell nuclei using UV light microscopy at 454 nm. Images were obtained with microscopy (200× magnification), and were measured using a Leica D6000 fluorescence microscope (Leica, Wetzlar, Germany).

### 5.14. Statistics

To perform statistical analyses, we used GraphPad Prism software version 6.0 (GraphPad Software Inc., San Diego, CA, USA). The three groups were compared with one-way ANOVA. Data are represented as the mean ± SD, and *p* < 0.05 was considered significant.

## Figures and Tables

**Figure 1 toxins-13-00745-f001:**
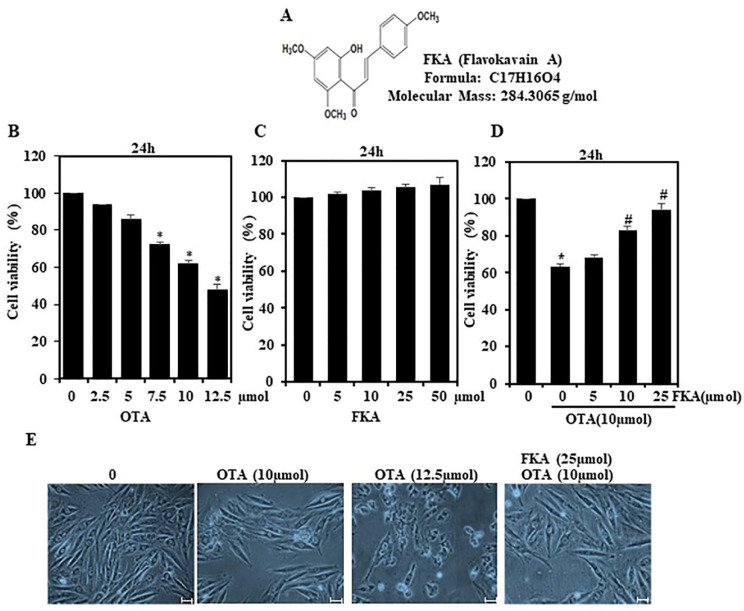
Effect of OTA and FKA on cell viability as calculated by MTT assay. (**A**) Chemical structure of FKA. (**B**) Cell viability was treated with different OTA concentrations (2.5–12.5 μmol) for 24 h. (**C**) Cell viability was determined with different FKA concentrations (5–50 μmol) for 24 h. Cells were pre-treated with FKA (5, 10, and 25 μmol for 2 h) and then stimulated with or without OTA (10 μmol) for 24 h. (**D**) Morphological changes in HUVECs. (**E**) Effect of FKA on OTA induced cells morphology for 24 h. * *p* < 0.05 denotes significant variations in comparison to control. # *p* < 0.05 denotes significant variations as compared to OTA alone and FKA with OTA treatment groups.

**Figure 2 toxins-13-00745-f002:**
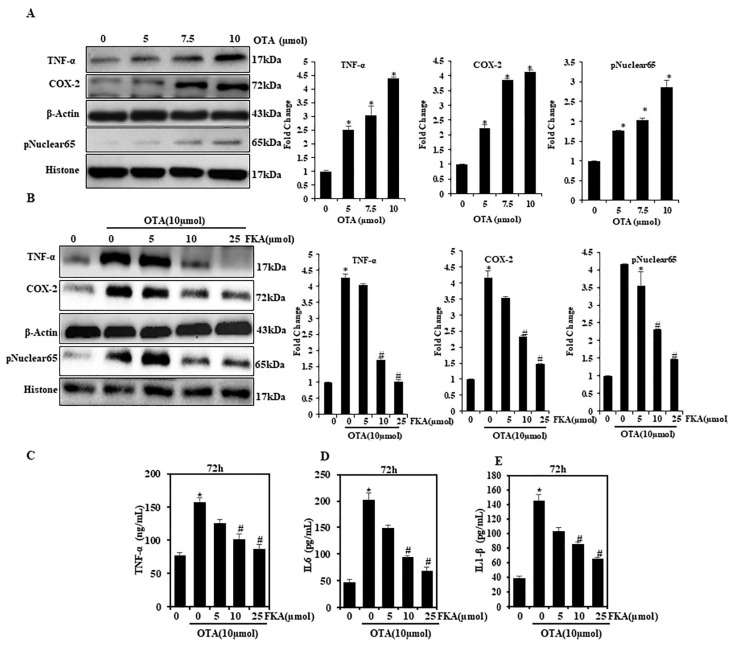
OTA-stimulated NFκB activation and pro-inflammatory cytokine expressions were suppressed in FKA-pretreated HUVECs. (**A**) HUVECs were treated with OTA (0, 5–10 μmol) for 24 h for the Western blot analysis of pNFκB (p65), COX-2, and TNF-α expression. (**B**) Pretreatment occurred with FKA (5–25 μmol) for 2  h and stimulated with OTA (10  μmol) for 24 h for the Western blot analysis of pNFκB (p65), COX-2, and TNF-α level. (**C**–**E**) The secretions of TNF-α, IL-6, and IL-1β were assayed with commercial ELISA kits. * *p* < 0.05 denotes significant variations compared with the control. # *p* < 0.05 denotes significant variations as compared to OTA alone and FKA with OTA treatment groups.

**Figure 3 toxins-13-00745-f003:**
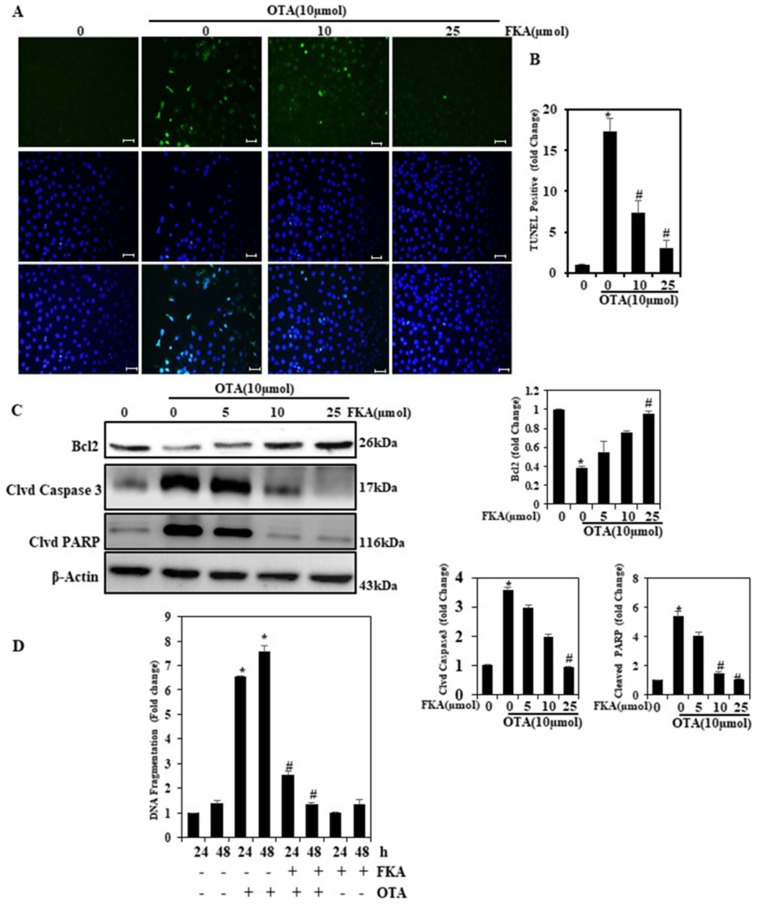
FKA inhibits apoptosis in endothelial cells. Cells were pretreated with FKA (0, 10 and 25 μmol for 2 h) followed by induction with OTA (10 μmol) for 24 h. (**A**) TUNEL assay. In the microscope fields (×400 magnification) the green fluorescence indicates the number of TUNEL-positive cells from three separate samples. (**B**) The fold change of apoptotic cells. (**C**) Bc-2, cleaved caspase-3, and PARP were monitored using Western blot analyses. Relative changes in Bcl-2, cleaved caspase-3, and PARP protein bands were measured by commercially available quantitative software (AlphaEase), with controls representing 1-fold. (**C**) Cells were pretreated with FKA (0, 10, and 25 μmol for 2 h) followed by induction with OTA (10 μmol) for the indicated times. (**D**) Apoptosis was measured by the degree of DNA fragmentation in the cytoplasm of cells treated with FKA. * *p* < 0.05 represents significant variations compared with the control. # *p* < 0.05 denotes significant variations as compared to OTA alone and FKA with OTA treatment groups.

**Figure 4 toxins-13-00745-f004:**
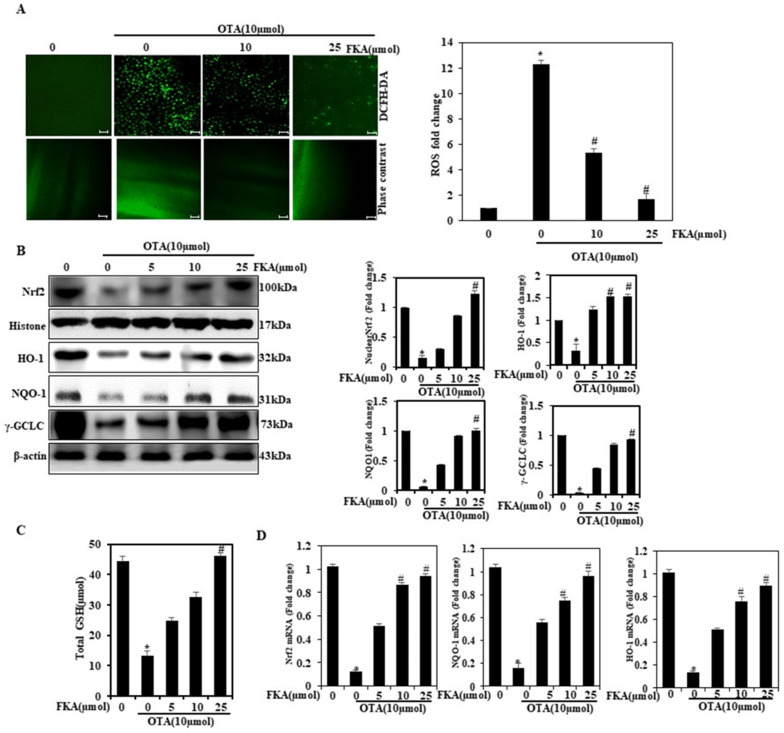
Effect of FKA on OTA-stimulated ROS levels, NQO-1, HO-1, and γ-GCLC antioxidant gene expression. (**A**) HUVECS were pretreated with FKA (0, 10, and 25 μmol for 2 h) followed by induction with OTA (10 μmol) for 24 h. The intracellular ROS levels were measured with flow cytometry method by the DCF fluorescence technique. (**B**) Nuclear Nrf2 expression levels were measured andNQO-1, HO-1, and γ-GCLC proteins were regulated using WB analysis. The expressions of NQO-1, HO-1, and γ-GCLC were measured with the WB method. (**C**) Concentration of intracellular GSH was assayed using a commercially available ELISA kit. (**D**) Nrf2, NQO-1, and HO-1 expression were analyzed by RT-PCR. * *p* < 0.05 denotes significant variations compared with the control. # *p* < 0.05 denotes significant variations as compared to OTA alone and FKA with OTA treatment groups.

**Figure 5 toxins-13-00745-f005:**
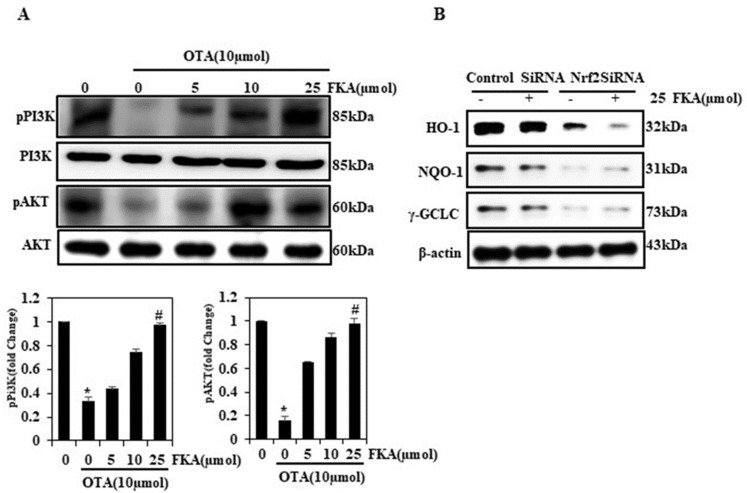
Effects of FKA and/or OTA on the PI3K/AKT pathways in HUVECs. Cells were pretreated with FKA (0, 10, and 25 μmol for 2  h) followed by induction with OTA (10  μmol) for 24  h. (**A**) Phosphorylation of PI3K and AKT was detected with Western blot. (**B**) Cells were transfected with siRNA against Nrf2 and/or a non-silencing control. The protein levels NQO-1, HO-1, and γ-GCLC were detected with Western blot analysis. * *p* < 0.05 denotes significant variations compared with the control. # *p* < 0.05 denotes significant variations as compared to OTA alone and FKA with OTA treatment groups.

**Figure 6 toxins-13-00745-f006:**
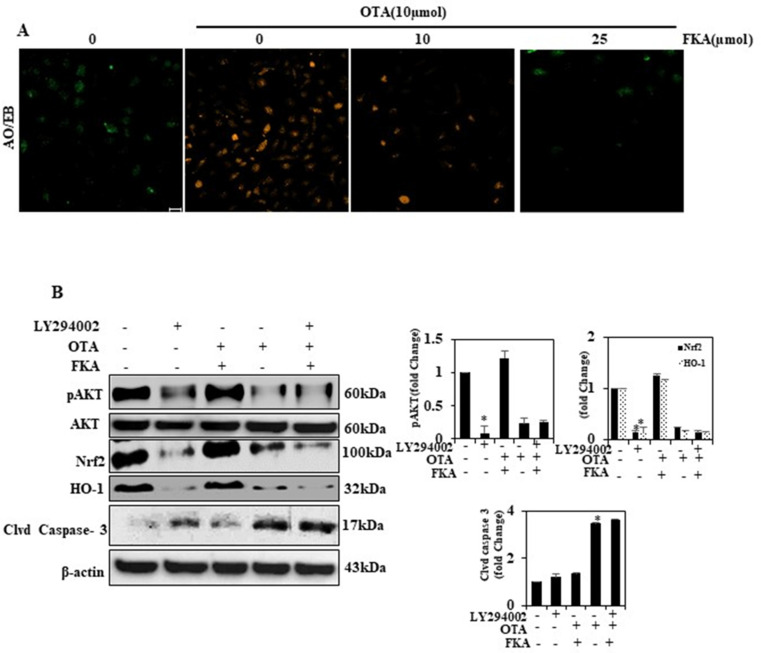
FKA mediates PI3K/AKT activation. Cells were pretreated with FKA (0, 10, and 25 μmol for 2 h) followed by induction with OTA (10 μmol) for 24 h. (**A**) Cell apoptosis observed by acridine orange/ethidium bromide (AO/EB) staining. (**B**) Cells were pretreated with LY294002, (30 μM) for 2 h, and FKA (25 μmol) and/or OTA (10 μmol) for 24 h to determine the pAkt, Nrf2, and HO-1 levels. * *p* < 0.05 denotes significant variations compared with the control.

## Data Availability

The data that support the findings of this study are available from the corresponding author upon reasonable request.

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
