# Peer review of "Anti-Apoptotic Effect of Flavokawain A on Ochratoxin-A-Induced Endothelial Cell Injury by Attenuation of Oxidative Stress via PI3K/AKT-Mediated Nrf2 Signaling Cascade"

_toxins, 2021, doi:10.3390/toxins13110745_

Round 1

Reviewer 1 Report

Please supplement the discussion on how phosphorylated NRF2 changes as a result of FKA treatment.

I suggest that the spellings and typos be corrected.

Author Response

Reviewer 1

Please supplement the discussion on how phosphorylated NRF2 changes as a result of FKA treatment.

Thank you for your valuable comments. We have revised the manuscript as per your suggestions.

I suggest that the spellings and typos be corrected.

The typo errors and spelling have been corrected as per your suggestions.

Thank you for all your valuable comments and questions, which allowed us to improve the quality of the manuscript.

If any responses are unclear or you wish additional changes, please do not hesitate to let us know.

Reviewer 2 Report

The study examined effects of flavokawain A (FKA) against ochratoxin A (OTA)-induced oxidative stress in HUVEC cells. Although results are interesting and many methods have been employed to reveal the molecular mechanisms of FKA action, the manuscript has too many serious flaws and cannot be accepted in the present form. The authors are suggested to carefully read the manuscript and correct all inconsistencies. In the present form, major concepts behind the study are not presented clearly. Professional assistance for English editing is also suggested. It is likely that some contradictions are consequence of language problems.

Here are some suggestions for the improvement:

Abstract – results along NFκB should be mentioned

Introduction is not focused - diverse pathologies were mentioned, from cancer to hepatotoxicity,  it is not clearly presented what is known about oxidative stress, endothelial dysfunction, ochratoxin A, and FKA. Introduction should be re-written and focused on 3 major topics: endothelial dysfunction and oxidative stress, ochratoxin A and pathology of endothelial cells, previously known effects of FKA on endothelial cells or some known mechanisms of FKA action from the literature.

Discussion – Results are mostly repeated instead of discussed

Figure 1:

MTT results (1D) do not match with morphological changes: for OTA (10 µM) it seems that viability is much less than 60% on photograph; OTA (25 µM) – it is not clear what this photograph represent; although for control and FKA + OTA viability is app. 100%, cell density and morphological appearance are different between these two groups

Figure 1B – marks of statistical significance should be added

Magnification should be added

Figure 2:

Densitometry, together with deviations and mark of significance, should be added for TNF-α, COX-2 and pNuclear65

If the numbers above bands represent intensities, results should be explained.  In the present form, it is not clear what these numbers mean, and they are confusing (some bands do not have numbers; numbers above β-actin and histone indicate that their expressions are quite variable between groups)

It seems that 5 µM FKA + OTA stimulates NFκB expression (2B) -please comment this increase

According to line 120, WB of IL-1β is missing

Figure 3:

Figure 3A – bottom photographs (phase contrast) should be explained, photographs of cells should be represented instead of green pictures

Densitometry of ROS levels should be added

Densitometry of γ-GCLC expression barely matches with the representative blot

Line 166 – the authors wrote “We measured the data as the fold over basal levels of the expression of antioxidant proteins at varying time points” – varying time points are not represented

Line 184 – increase in γ-GCLC is likely involved in GSH increase (in addition to ROS scavenging)

Figure 4:

Line 208 – the authors explained – “Data obtained from Western blot demonstrated that FKA increased the HO-1, γ-GCLC and NQO-1 expressions in the control cells, contrary to the Nrf2 knockdown cells (Fig 4 B)” – no increase is evident on Figure 4B

Below the title Nrf2 siRNA expression of HO-1 is decreased, whereas expressions of γ-GCLC and NQO-1 are increased – the expressional changes of genes under Nrf2 control are not in the same direction – conclusions are questionable; overall - results of silencing are not explained in an understandable manner, control group is missing

Line 215 – “FKA generated significant 5-and 7-fold decreases in DNA fragmentation in HUVECs (Fig. 4C), following the treatment for 24 h and 48 h, respectively, in comparison to the cells treated with the combination of FKA and OTA.”  - probably it should be in comparison to the cells treated with the OTA?

DNA fragmentation that is marker of apoptosis should be part of Figure 6

Figure 5:

Phase contrast photographs should be represented together with AO/EB pictures

Number of EB stained cells for 10 µM FKA + 10 µM OTA is in contrast with the viability represented in Figure 1D

Line 233 – “As shown in Figure 5B, cells pretreated with the combination of OTA and FKA had upregulated pAKT, Nrf2, and HO-1in comparison to control cells.” - Control group (without any treatment) is missing from Figure 5B

Line 249 – “The qRT-PCR results showed significant upregulation of the proteins” – mRNA, not proteins

Results of qRT-PCR analysis should be part of Figure 3 (together with the WB analysis of protein expression)

Other suggestions:

Line 90: the authors wrote “revealing the primary action of Nrf2 signaling on antioxidant pathways.” – Nrf2 is not focus of the study, it should be written how FKA affects Nrf2 pathway

Line 67 - "Numerous studies have investigated the endothelial protective effects of natural compounds against liver injury" – confusing

Line 103 – “Moreover, it prevents the increased growth induced by TGF-β1” – unclear, not related to this study, copied from some other study?

Line 205 – “2.7. Knockdown of Nrf2 (siRNA) Suppresses FKA-Induced Oxidative Stress of HUVECs” – FKA did not induce oxidative stress, correct the title

Line 330 – “The findings showed that FKA inhibited PI3K/AKT and Nrf2 expressions” – not correct

Line 176- “Concentration of intracellular GSH was assayed using a commercially available ELISA kit.” – in contrast to method described in lines 373-379, it is not clear how is GSH measured

Line 387 – primary and secondary antibodies, as well as their dilutions, are not indicated

Line 471 – overstated, endothelium function is not monitored

Line 158- Keap-1/Nrf2/ARE is primarily an antioxidative pathway

Lines 135-137: “ROS are detrimental; ROS in small concentration could contribute to redox signaling that may facilitate normal functions of cells, adaptation, and the prevention of diseases” – confusing, should be explained better

Line 17: chalcone- why is this term used only in abstract?

Line 29 - neutrophil recruitment of neutrophil?

Lines 26-28: similar to www.angioproteomie.com/commerce/ccc2491-animal-endothelial-cells.htm

Line 31 – ref. 3 is not appropriate

Line 32 - …”toxins could either be pathogenic or saprophytic” - ?

Lines 37-38 similar to lines 40-41 – delete duplication

Line 46: consequences instead of causes

Line 45 similar to line 48 – delete duplication

Line 60. „According to a large number of studies” – please insert references

Line 308 – PDK?

Author Response

The study examined effects of flavokawain A (FKA) against ochratoxin A (OTA)-induced oxidative stress in HUVEC cells. Although results are interesting and many methods have been employed to reveal the molecular mechanisms of FKA action, the manuscript has too many serious flaws and cannot be accepted in the present form. The authors are suggested to carefully read the manuscript and correct all inconsistencies. In the present form, major concepts behind the study are not presented clearly. Professional assistance for English editing is also suggested. It is likely that some contradictions are consequence of language problems.

Thank you for your valuable comments.

We corrected our revised manuscript using English editing with MDPI. Thank you for your valuable comments.

Here are some suggestions for the improvement:

Abstract – results along NFκB should be mentioned

Thank you for your valuable comments. In Our revised manuscript we have mentioned about NFкB.

Introduction is not focused - diverse pathologies were mentioned, from cancer to hepatotoxicity,  it is not clearly presented what is known about oxidative stress, endothelial dysfunction, ochratoxin A, and FKA. Introduction should be re-written and focused on 3 major topics: endothelial dysfunction and oxidative stress, ochratoxin A and pathology of endothelial cells, previously known effects of FKA on endothelial cells or some known mechanisms of FKA action from the literature.

Thank you for your valuable comments. We have revised our manuscript as per your suggestions.

Discussion – Results are mostly repeated instead of discussed

 Thank you for your valuable comments. We have modified our revised manuscript as per your suggestions.

Figure 1:

MTT results (1D) do not match with morphological changes: for OTA (10 µM) it seems that viability is much less than 60% on photograph; OTA (25 µM) – it is not clear what this photograph represent; although for control and FKA + OTA viability is app. 100%, cell density and morphological appearance are different between these two groups.

Thank you for your comments. In the revised manuscript,  we have added a clear image. Thank you.

Figure 1B – marks of statistical significance should be added

We have made the changed as per your suggestions.

Magnification should be added

 We have added the magnification in our revised manuscript as per your suggestions.

Figure 2:

Densitometry, together with deviations and mark of significance, should be added for TNF-α, COX-2 and pNuclear65

We have made the modifications in our revised manuscript as per your suggestions.

If the numbers above bands represent intensities, results should be explained.  In the present form, it is not clear what these numbers mean, and they are confusing (some bands do not have numbers; numbers above β-actin and histone indicate that their expressions are quite variable between groups)

It seems that 5 µM FKA + OTA stimulates NFκB expression (2B) -please comment this increase

We are extremely sorry for that loading error. In future studies, we shall take care of this. Thank you for your comments

According to line 120, WB of IL-1β is missing

 It was a Typographic error. Now we have included as per your suggestions.

Figure 3:

Figure 3A – bottom photographs (phase contrast) should be explained, photographs of cells should be represented instead of green pictures

This phase contrast was used to investigate any structural changes.

Densitometry of ROS levels should be added

We have done the modifications in the revised manuscript as per your suggestions.

Densitometry of γ-GCLC expression barely matches with the representative blot

In our revised manuscript we have corrected images (contrast of the picture). We shall take care in our future studies.

Line 166 – the authors wrote “We measured the data as the fold over basal levels of the expression of antioxidant proteins at varying time points” – varying time points are not represented

It was typographic error. We have included as per your suggestions.

Line 184 – increase in γ-GCLC is likely involved in GSH increase (in addition to ROS scavenging)

We have made the changes in our revised manuscript as per your suggestions.

Figure 4:

Line 208 – the authors explained – “Data obtained from Western blot demonstrated that FKA increased the HO-1, γ-GCLC and NQO-1 expressions in the control cells, contrary to the Nrf2 knockdown cells (Fig 4 B)” – no increase is evident on Figure 4B

In our revised manuscript,  we have corrected the images (contrast of the picture). We shall care about this in our future studies.

Below the title Nrf2 siRNA expression of HO-1 is decreased, whereas expressions of γ-GCLC and NQO-1 are increased – the expressional changes of genes under Nrf2 control are not in the same direction – conclusions are questionable; overall - results of silencing are not explained in an understandable manner, control group is missing

It was typographic error. Now we have included as per your suggestions.

Line 215 – “FKA generated significant 5-and 7-fold decreases in DNA fragmentation in HUVECs (Fig. 4C), following the treatment for 24 h and 48 h, respectively, in comparison to the cells treated with the combination of FKA and OTA.”  - probably it should be in comparison to the cells treated with the OTA?

 In our revised manuscript, we have modified as per your suggestions.

DNA fragmentation that is marker of apoptosis should be part of Figure 6

 In our revised manuscript, we have modified as per your suggestions.

Figure 5:

Phase contrast photographs should be represented together with AO/EB pictures

We are extremely sorry as we did not take photo for phase contrast. Thank you for your valuable comments. For  future studies, we shall  consider it.

Number of EB stained cells for 10 µM FKA + 10 µM OTA is in contrast with the viability represented in Figure 1D

We had put a clear image in our revised manuscript.

Line 233 – “As shown in Figure 5B, cells pretreated with the combination of OTA and FKA had upregulated pAKT, Nrf2, and HO-1in comparison to control cells.” - Control group (without any treatment) is missing from Figure 5B

It was typographic error. We have included as per your suggestions.

Line 249 – “The qRT-PCR results showed significant upregulation of the proteins” – mRNA, not proteins

It was typographic error. We have included as per your suggestions.

Results of qRT-PCR analysis should be part of Figure 3 (together with the WB analysis of protein expression)

In our revised manuscript we have modified as per your suggestions.

Other suggestions:

Line 90: the authors wrote “revealing the primary action of Nrf2 signaling on antioxidant pathways.” – Nrf2 is not focus of the study, it should be written how FKA affects Nrf2 pathway

 In our revised manuscript we have modified as per your suggestions.

Line 67 - "Numerous studies have investigated the endothelial protective effects of natural compounds against liver injury" – confusing

  In our revised manuscript we have modified as per your suggestions.

Line 103 – “Moreover, it prevents the increased growth induced by TGF-β1” – unclear, not related to this study, copied from some other study?

    In our revised manuscript we have modified as per your suggestions.

Line 205 – “2.7. Knockdown of Nrf2 (siRNA) Suppresses FKA-Induced Oxidative Stress of HUVECs” – FKA did not induce oxidative stress, correct the title

      In our revised manuscript we have modified as per your suggestions.

Line 330 – “The findings showed that FKA inhibited PI3K/AKT and Nrf2 expressions” – not correct

  In our revised manuscript we have modified as per your suggestions.

Line 176- “Concentration of intracellular GSH was assayed using a commercially available ELISA kit.” – in contrast to method described in lines 373-379, it is not clear how is GSH measured

In our revised manuscript we have modified in our methods section as per your suggestions.

Line 387 – primary and secondary antibodies, as well as their dilutions, are not indicated

If we include the names of antibodies, there appears more similarities in the article.  We wanted to avoid similarities in the article. In the revised manuscript we have included antibodies dilutions.

Line 471 – overstated, endothelium function is not monitored

  Our revised manuscript we modified as suggested you.

Line 158- Keap-1/Nrf2/ARE is primarily an antioxidative pathway

In our revised manuscript we have modified as per your suggestions.

Lines 135-137: “ROS are detrimental; ROS in small concentration could contribute to redox signaling that may facilitate normal functions of cells, adaptation, and the prevention of diseases” – confusing, should be explained better

 We have corrected it in our revised manuscript.

Line 17: chalcone- why is this term used only in abstract?

 We have made the changes in our revised manuscript.

Line 29 - neutrophil recruitment of neutrophil?

We have made the changes in our revised manuscript.

Lines 26-28: similar to www.angioproteomie.com/commerce/ccc2491-animal-endothelial-cells.htm

We have made the changes in our revised manuscript.

Line 31 – ref. 3 is not appropriate

We have made the changes in our revised manuscript.

Line 32 - …”toxins could either be pathogenic or saprophytic” - ?

We have made the changes in our revised manuscript.

Lines 37-38 similar to lines 40-41 – delete duplication

We have made the changes in our revised manuscript.

Line 46: consequences instead of causes

We have made the changes in our revised manuscript.

Line 45 similar to line 48 – delete duplication

We have made the changes in our revised manuscript.

Line 60. „According to a large number of studies” – please insert references

We have made the changes in our revised manuscript.

Line 308 – PDK?

We have made the changes in our revised manuscript.

Thank you for all your valuable comments and questions, which allowed us to improve the quality of the manuscript.

If any responses are unclear or you wish additional changes, please do not hesitate to let us know.

Reviewer 3 Report

Dear Authors, I have some comments for you manuscript:

  1. Latin should be italicized
  2. There are some typos and absence of spaces.
  3. All figures are very crowded, please provide sufficient distance between sections A, B, C, etc
  4. The scale on images from microscope is absent everywhere
  5. The quality of all figures is too low, please provide higher quality

Author Response

Dear Authors, I have some comments for you manuscript:

  1. Latin should be italicized

Thank you for your valuable comments. We revised our manuscript as per your suggestions.

  1. There are some typos and absence of spaces.

We corrected our revised manuscript using English editing with MDPI. Thank you for your valuable comments.

  1. All figures are very crowded, please provide sufficient distance between sections A, B, C, etc

Our revised manuscript was modified as per your suggestions.

  1. The scale on images from microscope is absent everywhere

Our revised manuscript was modified as per your suggestions.

  1. The quality of all figures is too low, please provide higher quality

We improved the quality of the figures in the revised manuscript.

Thank you for all your valuable comments and questions, which allowed us to improve the quality of the manuscript.

If any responses are unclear or you wish additional changes, please do not hesitate to let us know.

Reviewer 4 Report

In the manuscript titled “Anti-apoptotic effect of Flavokawain A on Ochratoxin A-induced endothelial cell injury by attenuation of oxidative stress via PI3K/AKT mediated Nrf2 signaling cascade”, the authors found that flavokawain A (FKA) inhibits ochratoxin A (OTA)-induced cell death and the production of inflammatory cytokines in HUVECs. Furthermore, the authors showed that the inhibitory effects of FKA is due to the activation of the PI3K/AKT/Nrf2 signaling pathway. The results are generally sound. However, the authors fail to rule out the possibility that FKA counteracts the toxic effects of OTA by interacting directly with it. In addition, I think the order of the figures needs to be rearranged.

Major comments

  1. I think it would be better to put Fig. 6 after Fig. 2.
  2. There are several experiments on apoptosis in several figures, but I don’t think Fig. 4A and Fig. 5C are necessary.
  3. In Fig. 4, the authors should show that whether Nrf2 siRNA is canceled the inhibitory effect of FKA on OTA-induced cell death in HUVECs. In addition, I think Fig.4A should be moved to Fig. 5.
  4. Similarly, In Fig. 5, the authors should show that whether LY294002 is canceled the inhibitory effect of FKA on OTA-induced cell death in HUVECs. In addition, I think Fig.5C should be moved to Fig. 4.

Mainor comments

  1. In Fig. 1A, the authors need to improve the quality of the chemical structural formulas.
  2. In Fig. 2A and B, the authors need to describe the treatment time of OTA.
  3. The legend of Fig. 3A states that the ROS levels were measured by flow cytometry, but what is shown is a micrograph.

Author Response

Reviewer 4

In the manuscript titled “Anti-apoptotic effect of Flavokawain A on Ochratoxin A-induced endothelial cell injury by attenuation of oxidative stress via PI3K/AKT mediated Nrf2 signaling cascade”, the authors found that flavokawain A (FKA) inhibits ochratoxin A (OTA)-induced cell death and the production of inflammatory cytokines in HUVECs. Furthermore, the authors showed that the inhibitory effects of FKA is due to the activation of the PI3K/AKT/Nrf2 signaling pathway. The results are generally sound. However, the authors fail to rule out the possibility that FKA counteracts the toxic effects of OTA by interacting directly with it. In addition, I think the order of the figures needs to be rearranged.

 Major comments

  1. I think it would be better to put Fig. 6 after Fig. 2.

We have rearranged the figure in our revised manuscript as per your suggestions.

  1. There are several experiments on apoptosis in several figures, but I don’t think Fig. 4A and Fig. 5C are necessary.

This is first study for apoptosis induced by OTA with HUVEC cells. We want more confirmation as whether FKA inhibit the apoptosis or not. 

  1. In Fig. 4, the authors should show that whether Nrf2 siRNA is canceled the inhibitory effect of FKA on OTA-induced cell death in HUVECs. In addition, I think Fig.4A should be moved to Fig. 5.

Reviewer 2 suggested to rearrange this figure and we have already rearranged. Thank you for your valuable comments.

  1. Similarly, In Fig. 5, the authors should show that whether LY294002 is canceled the inhibitory effect of FKA on OTA-induced cell death in HUVECs. In addition, I think Fig.5C should be moved to Fig. 4.

Reviewer 2 suggested to rearrange this figure and we have already rearranged. Thank you for your valuable comments.

Maior comments

  1. In Fig. 1A, the authors need to improve the quality of the chemical structural formulas.

We have done the changes in the revised manuscript as per your suggestions.

  1. In Fig. 2A and B, the authors need to describe the treatment time of OTA.

We have included in the figure legends in our revised manuscript. Thank you for your valuable comments.

  1. The legend of Fig. 3A states that the ROS levels were measured by flow cytometry, but what is shown is a micrograph.

We have included ROS measuring graph in our revised manuscript.  Thank you for your valuable comments.

Thank you for all your valuable comments and questions, which allowed us to improve the quality of the manuscript.

If any responses are unclear or you wish additional changes, please do not hesitate to let us know.

Round 2

Reviewer 2 Report

Introduction is improved and could be considered satisfactory. Other parts of the manuscript are also improved in many elements. However, some questions still remain open and should be discussed.

Figure legend of Figure 1 is not correct (from A-D is included, E is missing). Description of 1E should be explained in details. It is not clear why photograph of cells exposed to OTA 25 μM (all other experiments are performed with 10 μM OTA) is included in Figure 1, while FKA 25 μM is not. It seems to me that different type of cells is now represented in Figure 1E (in comparison with the original manuscript) and that these cells morphologically do not match HUVEC cells. This needs to be clarified (perhaps their morphology varies during passaging?). The length of scale bars should be indicated.

In Figure 6B control group is missing. In line 320 the authors wrote „As shown in Figure 6B, cells pretreated with the combination of OTA and FKA upregulated pAKT, Nrf2, and HO-1. Comparison should be performed with control group that is not treated with LY294002 for which it is expected to downregulate pAkt expression. In the original version of the manuscript authors provided correct densitometric analysis that included all groups (cont, LY294002, OTA+FKA, OTA, OTA+FKA+ LY294002). It is not clear why this representation is omitted whereas the incomplete WB without the control group is left. WB and densitometric analysis of all 5 groups should be represented in Figure 6.

Line 131 – „we examined the effects of FKA when pre-treatment resulted in patterns of IL-1β and TNF-α expression“ – unclear, what is meant by patterns of IL-1β and TNF-α expression

Line 132 - „Western blot also demonstrated that OTA (10 μmol) stimulation overexpressed the COX-2, IL-1β, and TNF-α expressions (Fig 2B)“ – WB of IL-1β in Figure 2B is not represented

Line 255 – „FKA in scavenging excessive ROS induced by OTA by an increase in γ-GCLC is likely involved in the GSH increase in HUVECs (in addition to ROS scavenging)“ – not written clearly, my suggestion was related to the potential explanation of the observed GSH increase (GSH increase could be achieved via FKA-mediated direct ROS scavenging and through the upregulation of γ-GCLC that is involved in GSH synthesis)

Line 72: reference cited is not related to hepatic injury

Author Response

Reviewer 2

Figure legend of Figure 1 is not correct (from A-D is included, E is missing). Description of 1E should be explained in details.

Thank you, have done as suggested you, page line 118

It is not clear why photograph of cells exposed to OTA 25 μM (all other experiments are performed with 10 μM OTA) is included in Figure 1, while FKA 25 μM is not. It seems to me that different type of cells is now represented in Figure 1E (in comparison with the original manuscript) and that these cells morphologically do not match HUVEC cells. This needs to be clarified (perhaps their morphology varies during passaging?). The length of scale bars should be indicated.

Thank you for your valuable comments

Actual dose 12.5 µM, typographic error, revised manuscript we have correct it.

The passage of cell line passage is different when compared with the original manuscript.  Therefore, the cells morphology changed.

In Figure 6B control group is missing. In line 320 the authors wrote „As shown in Figure 6B, cells pretreated with the combination of OTA and FKA upregulated pAKT, Nrf2, and HO-1. Comparison should be performed with control group that is not treated with LY294002 for which it is expected to downregulate pAkt expression. In the original version of the manuscript authors provided correct densitometric analysis that included all groups (cont, LY294002, OTA+FKA, OTA, OTA+FKA+ LY294002). It is not clear why this representation is omitted whereas the incomplete WB without the control group is left. WB and densitometric analysis of all 5 groups should be represented in Figure 6.

Our revised manuscript we have done this experiment with control group, and raw data submitted to editor.

Line 131 – „we examined the effects of FKA when pre-treatment resulted in patterns of IL-1β and TNF-α expression“ – unclear, what is meant by patterns of IL-1β and TNF-α expression

We have corrected this line in our revised manuscript

Line 132 - „Western blot also demonstrated that OTA (10 μmol) stimulation overexpressed the COX-2, IL-1β, and TNF-α expressions (Fig 2B)“ – WB of IL-1β in Figure 2B is not represented

typographic error, revised manuscript we have correct

Line 255 – „FKA in scavenging excessive ROS induced by OTA by an increase in γ-GCLC is likely involved in the GSH increase in HUVECs (in addition to ROS scavenging)“ – not written clearly, my suggestion was related to the potential explanation of the observed GSH increase (GSH increase could be achieved via FKA-mediated direct ROS scavenging and through the upregulation of γ-GCLC that is involved in GSH synthesis)

Thank you for your valuable comments we have corrected revised manuscript

Line 72: reference cited is not related to hepatic injury

Thank you, we have changed related reference

Rajendran, P., Ammar, R. B., Al-Saeedi, F. J., Mohamed, M. E., ElNaggar, M. A., Al-Ramadan, S. Y., Bekhet, G. M., and Soliman, A. M. (2021) Kaempferol inhibits zearalenone-induced oxidative stress and apoptosis via the PI3K/Akt-mediated Nrf2 signaling pathway: in vitro and in vivo studies. International Journal of Molecular Sciences 22, 217

Reviewer 4 Report

The authors have not fully complied with my request. They should show that whether the inhibition of the PI3K/Akt/Nrf2 axis canceled the inhibitory effect of FKA on OTA-induced cell death in HUVECs.

Author Response

Reviewer 4

The authors have not fully complied with my request. They should show that whether the inhibition of the PI3K/Akt/Nrf2 axis canceled the inhibitory effect of FKA on OTA-induced cell death in HUVECs.

We are sorry for forgetting to answer this question in the previous revised version of our manuscript. Now we have updated Fig 6 B, in which the western blot results of inhibitory effect of the PI3K/Akt/Nrf2 axis on the action of FKA on OTA-induced cell death in HUVECs are included.

Round 3

Reviewer 2 Report

Can be accepted.

Reviewer 4 Report

Thank you for responding to my request.